# Patterns of interactions among ICU interprofessional teams: A prospective patient-shift-level survey approach

**Hannah C. Ratliff**[1], **Olga Yakusheva**[1,2], **Emily M. Boltey**[3], **Deanna J. Marriott**[1], **Deena Kelly Costa**[4,5]*

1 School of Nursing, University of Michigan, Ann Arbor, MI, United States of America, 2 School of Public Health, University of Michigan, Ann Arbor, MI, United States of America, 3 VA Pittsburgh Healthcare System, Pittsburgh, PA, United States of America, 4 School of Nursing, Yale University, West Haven, CT, United States of America, 5 Section of Pulmonary, Critical Care & Sleep Medicine, School of Medicine, Yale University, New Haven, CT, United States of America

* deena.costa@yale.edu

## Abstract

### Background

The Awakening, Breathing Coordination, Delirium monitoring and Early mobility bundle (ABCDE) is associated with lower mortality for intensive care unit (ICU) patients. However, efforts to improve ABCDE are variably successful, possibly due to lack of clarity about who are the team members interacting when caring for each patient, each shift. Lack of patient shift-level information regarding who is interacting with whom limits the ability to tailor interventions to the specific ICU team to improve ABCDE.

### Objective

Determine the number and types of individuals (i.e., clinicians and family members) interacting in the care of mechanically ventilated (MV) patients, as reported by the patients' assigned physician, nurse, and respiratory therapist (RT) each shift, using a network science lens.

### Methods

We conducted a prospective, patient-shift-level survey in 2 medical ICUs. For each patient, we surveyed the assigned physician, nurse, and RT each day and night shift about who they interacted with when providing ABCDE for each patient-shift. We determined the number and types of interactions, reported by physicians, nurses, and RTs and day versus night shift.

### Results

From 1558 surveys from 404 clinicians who cared for 169 patients over 166 shifts (65% response rate), clinicians reported interacting with 2.6 individuals each shift (physicians: 2.65, nurses: 3.33, RTs: 1.86); this was fewer on night shift compared to day shift (1.99 versus 3.02). Most frequent interactions were with the bedside nurse, attending, resident,

**Data Availability Statement:** Data were collected from participants in two sites and contain potentially identifying patient and participant (clinician) information. Due to the small sample

size of the study and institutional restrictions on sharing of data when participants (clinicians or patients) could be identifiable, according to the University of Michigan IRB, the authors cannot make these data publicly available. However, individuals interested in acquiring these data for reproducibility purposes can contact the University of Michigan Institutional Review Board at irbmed@umich.edu or 734.763.4768."

**Funding:** Funding: This study was funded in part by the Agency for Healthcare Research and Quality (K08HS024552 PI Costa, www.ahrq.gov), University of Michigan School of Nursing Donor Funding (MPIs Costa & Yakusheva; www.nursing.umich.edu) and the Sigma Theta Tau Society Rho Chapter Research grant (PI Ratliff, https://nursing.umich.edu/about/nursing-michigan/sigma-rho-chapter). The funders had no role in the development and design of the study, interpretation of findings, decision to publish or preparation of the manuscript.

**Competing interests:** Authors have declared that no competing interests exist.

**Abbreviations:** ABCDE, the Awakening, Breathing Coordination, Delirium and Early Mobility bundle; ABCDEF, the Awakening, Breathing, Choice of Drugs, Delirium monitoring and management, Early mobility, and Family engagement bundle; EHR, electronic health record; ICU, intensive care unit; MV, mechanically ventilated; RTs, respiratory therapists.

intern, and RT; family member interactions were reported in less than 1 in 5 surveys (12.2% of physician surveys, 19.7% of nurse surveys, 4.9% of RT surveys).

## Interpretation

Clinicians reported interacting with 3–4 clinicians each shift, and fewer on nights. Nurses interacted with the most clincians and family members. Interventions targeting shift-level teams, focusing on nurses and family members, may be a way to improve ABCDE delivery and ICU teamwork.

## Introduction

High quality mechanical ventilation care, such as the Awakening, Breathing, Coordination, Delirium and Early Mobility (ABCDE) bundle, has been associated with lower mortality for critically ill adults [1–3]. However, efforts to improve ABCDE delivery have been variably successful [4]. A study conducted in 2015–2017 in 68 intensive care units (ICUs) across the United States found ABCDE performance was only 4%; after receiving an intervention to improve ABCDE, performance only increased to 12% [4]. Recent data suggests that ABCDE delivery is much lower as a result of the COVID-19 pandemic [5, 6], suggesting opportunity for improvement.

Delivery of the ABCDE bundle is a complex team practice [7], with data suggesting that the nurse, physician, and respiratory therapist are the clinicians involved [8]. Delivering ABCDE involves clinicians coordinating interdependent tasks like awakening the patient, coordinating extubation, assessing for delirium and encouraging mobility [9]. As such, ABCDE involves communication, coordination, and interaction among the interprofessional team [7, 10, 11], which are also foundational processes for effective teamwork [12]. Indeed, prior work has found that interactions among team members are integral to teamwork [13] and improved patient outcomes [14, 15]. Despite this recognition, there is limited empirical evidence about team member interactions when delivering ABCDE to each patient, each shift [16]. By examining interactions among the team members who deliver ABCDE, we can identify areas for possible intervention that have not been studied before, as a way to improve ABCDE delivery and teamwork.

Building off standard network science approaches that survey individuals to identify who interacts with whom (e.g., to whom they provide and receive information [17–19]), we conducted a longitudinal prospective shift-level survey of ICU clinicians caring for mechanically ventilated adults in 2 ICUs. The purpose of this study was to determine the number and types of individuals (i.e., clinicians and family members) interacting in the care of mechanically ventilated patients (i.e., ABCDE bundle) each shift, as reported by the patient's assigned physician, nurse, and respiratory therapist (RT), in ICUs. The goal is to inform interventions to improve the interprofessional team's ability to deliver ABCDE to each patient, each shift.

## Methods

### Study design

We conducted a prospective in-person patient-shift-level survey of the nurses, physicians and respiratory therapists caring for each patient each shift in two study ICUs; we asked each clinician to identify all individuals they interacted with each shift, for a specific patient's ABCDE care.

When determining our study design, we made two purposeful decisions. First, we focused on interactions among team members about the ABCDE bundle [17, 18, 20]. ABCDE is an

evidence-based, multidisciplinary care bundle that is associated with decreased lengths of stay, duration of mechanical ventilation, prevalence and duration of delirium, and decreased health care costs for adult MV patients [1–3, 21]. The ABCDE bundle consists of individual bundle components: Awakening (i.e, spontaneous awakening trials), Breathing (i.e. spontaneous breathing trials), Coordination (i.e, coordination of A&B to facilitate extubation), Delirium monitoring and Early mobility [1, 22]. Thus, at a minimum, we would expect 4–5 interactions among the team each shift, corresponding to each respective bundle component (i.e., one interaction for each: A, B, C, D and E). Second, we surveyed the nurse, physician, and RT assigned to each patient each shift because our prior work demonstrated these are the key clinicians involved in mechanical ventilation care [8] and there was substantial variation in ICU staffing for allied health professionals [23–26] in our study sites; specifically, our study units did not have pharmacists or physical therapists routinely present in these units. Given allied health staffing variation, our in-person survey design, the need for adequate response rates, and the need to compare results across units, we selected the nurses, physicians and RTs as our clinician respondents as they were routinely present in both units. Clinician respondents were able to indicate allied health professionals and any other clinician or family member they interacted with that shift for each MV patient (see Survey instrument description below).

## Setting and sample

Our study occurred in two 20-bed medical ICUs in the Midwest United States (one community hospital and one academic medical center). Each ICU had approximately 50–60 nurses employed, 5–10 respiratory therapists providing care primarily in the ICU and varying physician presence. For physician staffing, the academic medical ICU had 2 intensivst led teams that shared the patient workload; each team had at least 1 resident, one intern and a fellow. The community hospital ICU had two medical teams–one led by a advanced practice provider that cared for non-mechanically ventilated patients and another team led by an intensivist and included one resident. We chose these ICUs as study sites because prior MHA Keystone ICU survey data found that ABCDE was routine in these ICUs [10, 11].

To be included in the study, patients had to be 18 years and older, hospitalized in one of the study ICUs, and mechanically ventilated for at least 24 hours; a patient was included in the study multiple times if they met the inclusion criteria on data collection days. All physicians, nurses, and RTs who cared for the patients during the data collection days were included; other clinicians were not eligible as survey respondents.

Our study received approval from the Institutional Review Board (HUM00108315); we received verbal informed consent from all participants (and a waiver of documentation of informed consent).

## Survey instrument

Our survey asked respondends to report name, clinical role (nurse/physician/RT, date, time, and shift [day/night]). We then asked: 1) Who did you contact about ABCDE when caring for this specific mechanically ventilated patient? And 2) who contacted you about ABCDE when caring for this specific mechanically ventilated patient? The survey instructed respondents to interpret "contact" broadly, including communication during rounds and one-on-one interactions (e.g., with an nurse about spontaneous awakening trial). For each question, clinicians could select 11 individuals (attending/resident/fellow/intern/respiratory therapist/physical therapist/bedside nurse/charge nurse/nurse manager/other nurses/patient's family member), document their names, and list additional clinicians if applicable, including 'no one'.

We first piloted our survey in one study ICU for a week to determine preferred survey administration (paper versus electronic), timing of the survey, and to receive feedback. Participants provided feedback about survey flow, requested additional information about what sort of interactions we were interested in, indicated the paper survey was quickest (approximately 3 minutes or less) and the 0500 and 1400 administration times fit best with ICU workflow. See complete survey in the S1 File.

## Data collection

Data collection in ICU 1 occurred from June 1, 2017 –September 30, 2017 (12 weeks) and from January 1, 2018 –April 30, 2018 (16 weeks) in ICU 2. In each ICU, we collected data on an alternating schedule–one week collecting shift-level surveys and one week off—to minimize participant and researcher burden; many respondents filled out multiple surveys (since they cared for more than one patient each shift) and research assistants collected data every week-day. We chose a 1-week alternating schedule because MV patient's average duration of mechanical ventilation is approximately four days [27]. We administered a paper survey between 1300–1400 for day shifts and 0400–0500 for night shifts.

On each shift, we identified all patients who met inclusion criteria using the EHR, charge nurse assistance or using a whiteboard in each unit. Surveys were distributed to physicians (most often residents as they were the most available during our survey administration times), nurses, and RTs that were assigned to care for the patients that met inclusion criteria. Each survey was about a specific patient (i.e., patient's name and medical record number. Every day of data collection, our study team tracked patient bed number, physician, nurse, and RT assigned and if we received a completed survey from that clinician. We provided candy as an incentive; no monetary incentive was provided.

## Data analysis

We analyzed clincians' survey responses at the patient-shift level (i.e., a clinician's response to a survey about a specific patient during a specific shift); we used quantitative cross-sectional analyses for the full sample, and stratified by the type of reporting clinician (physician, nurse, RT) and by shift (day versus night). We calculated our response rate (total number of completed and returned surveys divided by the number of all surveys possible) and then calculated descriptive statistics for the full sample. We also calculated descriptive statistics by day versus night shift using numbers/percentages for dichotomous variables (e.g., a clinician checked/did not check interacting with the bedside nurse) and means/ranges for count variables (e.g., how many individuals does a clinician report interacting with over the patient's care)]. The results are presented in **Table 1**.

We examined the interprofessional pattern of interactions. We categorized survey responses by the type of interactions reported: 1) 'attending physician', 'resident physician', 'fellow' and 'intern' were categorized as physicians, 2) 'bedside nurse', 'charge nurse', 'nurse manager', and 'other nurse' were categorized as nurses, and 3) 'respiratory therapist' was left as its own category, RTs. We then calculated 3x3 interprofessional interaction matrixes showing the frequencies of each type of interprofessional interactions (physician, nurse, RT) for each type of reporting clinician (physician, nurse, RT). The matrixes were calculated for the full sample and by day versus night shift and tested for statistical differences between day and night shift interactions (see **Table 2**).

Lastly, we created frequency distributions for interactions with each type of individual in the set of all individuals involved in a patient's care during a shift (all physicians, nurses, RTs, other clinicians, and family members). We created three frequency distributions, one for each type of

Table 1. Descriptive statistics of the sample.

| | All shifts | Day Shifts | Night Shifts |
|---|---|---|---|
| Number of clinicians surveyed^: n | 404 | 263 | 172 |
| physician: n, % | 120, 29.7% | 86, 32.7% | 44, 25.6% |
| nurse: n, % | 217, 53.7% | 136, 51.71% | 102, 59.3% |
| RT: n, % | 67, 16.5% | 41, 15.59% | 26, 15.1% |
| Number of patients | 169 | 160 | 141 |
| Number of shifts | 166 | 94 | 72 |
| Number of patient-shifts: n | 802 | 474 | 328 |
| with 1 clinician surveyed: n, % | 255, 31.8% | 145, 30.6% | 110, 33.5% |
| with 2 clinicians surveyed: n, % | 344, 42.9% | 203, 42.8% | 141, 43% |
| with 3 clinicians surveyed: n, % | 197, 25.6% | 124, 26.2% | 73, 22.3% |
| with 4 clinicians surveyed: n, % | 6, 0.75% | 2, 0.42% | 4, 1.2% |
| Number of clinician surveys: n | 1,558 | 931 | 627 |
| physician: n, % | 499, 32.0% | 275, 29.5% | 224, 35.7% |
| nurse: n, % | 523, 33.6% | 337, 36.1% | 186, 29.7% |
| RT: n, % | 536, 35.0% | 319, 34.3% | 217, 34.6% |
| Number of reported individual interactions ∼ per survey: mean (min, max) | 2.60 (0, 9) | 3.02 (0,9) | 1.99 (0, 8) |
| physician: mean (min, max) | 2.65 (0, 8) | 3.44 (0, 8) | 1.66 (0, 8) |
| nurse: mean (min, max) | 3.33 (0, 9) | 3.64 (0, 9) | 2.75 (0, 8) |
| RT: mean (min, max) | 1.86 (0, 7) | 1.98 (0, 5) | 1.67 (0, 7) |

^ "Surveyed clinicians" refers to the clinicians (MDs, RNs, and RTs) who completed surveys in the study.

∼ "Referenced individuals" refers to those individuals (clinicians, family members) who were reported as a clinician contact that shift, in clinician surveys.

MD = physician; RN = registered nurse; RT = respiratory therapist

reporting clinician (physician, nurse, RT). The histograms for the full sample are presented in **Figs 1–3**; similar histograms for day versus night shift analyses are presented in S1–S3 Figs.

## Results

We received 1558 complete surveys (499 physician surveys, 523 nurse surveys, and 536 RT surveys) from 404 clinicians who cared for 169 patients over 166 shifts for a total response rate of 65% (Table 1). We collected more surveys during the day shift than the night shift (Table 1). The number of clinicians surveyed per patient-shift varied between 1 and 4, with the majority of patient shifts (42.9%) having 2 clinicians surveyed. Physicians, nurses, and RTs were equally represented among surveyed respondents (i.e., about one-third from each clinician category).

The average number of interactions within a patient's ICU team was 2.60 (range 0–9; Table 1). Within each team, physicians reported an average of 2.65 interactions (range 0–8), nurses reported an average of 3.33 interactions (range 0–9), and RTs reported 1.86 interactions (range 0–7). Across all shifts, the reported interactions were fewer on night shift (1.99, range 0–8) compared to day shift (3.02, range 0–9). The largest difference in interactions on night shift, compared to day shift, was reported by physicians (3.44, range 0–9 on day shift compared to 1.66, range 0–8 on night shift); RTs reported a similar number of interactions across both shifts (1.67, range 0–7 on night shift compared to 1.98, range 0–5, on day shift).

Most interactions each shift were interprofessional (Table 2). Physicians reported the highest frequency of interactions with nurses: 70.5% of physician surveys reported interacting with at least one nurse (e.g., a bedside nurse or a charge nurse), followed by other physicians (57.3%

**Table 2. Distribution of clinician interactions by clinician type, day, and night shift.**

|  | All shifts | Day shifts | Night shifts | p-value |
|---|---|---|---|---|
| Total number of physician surveys | 499 | 275 | 224 |  |
| Reported physician: n, % | 286, 57.3% | 205, 74.5% | 81, 25.6% | <0.001 |
| Reported nurse: n, % | 352, 70.5% | 221, 80.4% | 131, 59.3% | <0.001 |
| Reported RT: n, % | 241, 48.3% | 168, 61.2% | 73, 15.1% | <0.001 |
| Total number of nurse surveys | 523 | 337 | 186 |  |
| Reported physician: n, % | 412, 78.8% | 281, 83.3% | 131,70.4% | 0.0005 |
| Reported nurse: n, % | 195, 37.3% | 112, 33.2% | 83, 44.6% | 0.009 |
| Reported RT: n, % | 380, 72.7% | 236, 70.0% | 144, 77.4% | 0.070 |
| Total number of RT surveys | 536 | 319 | 217 |  |
| Reported physician: n, % | 278, 51.9% | 171, 53.6% | 107, 49.3% | 0.328 |
| Reported d nurse: n, % | 417, 77.8% | 253, 79.3% | 164, 75.6% | 0.307 |
| Reported RT: n, % | 62, 11.6% | 45, 14.1% | 17, 7.8% | 0.026 |

of surveys), and RTs (48.3% of surveys). Nurses also reported the highest frequency of interactions with physicians (78.8% of surveys), followed by RTs (72.7% of surveys), and other nurses (37.3% of surveys). RTs reported the highest frequency of interactions with nurses (77.8% of surveys), followed by physicians (51.9% of surveys) and other RTs (11.6%). For nearly all nurse and physician surveys, the differences between day and night shift interactions were statistically significant (Table 2).

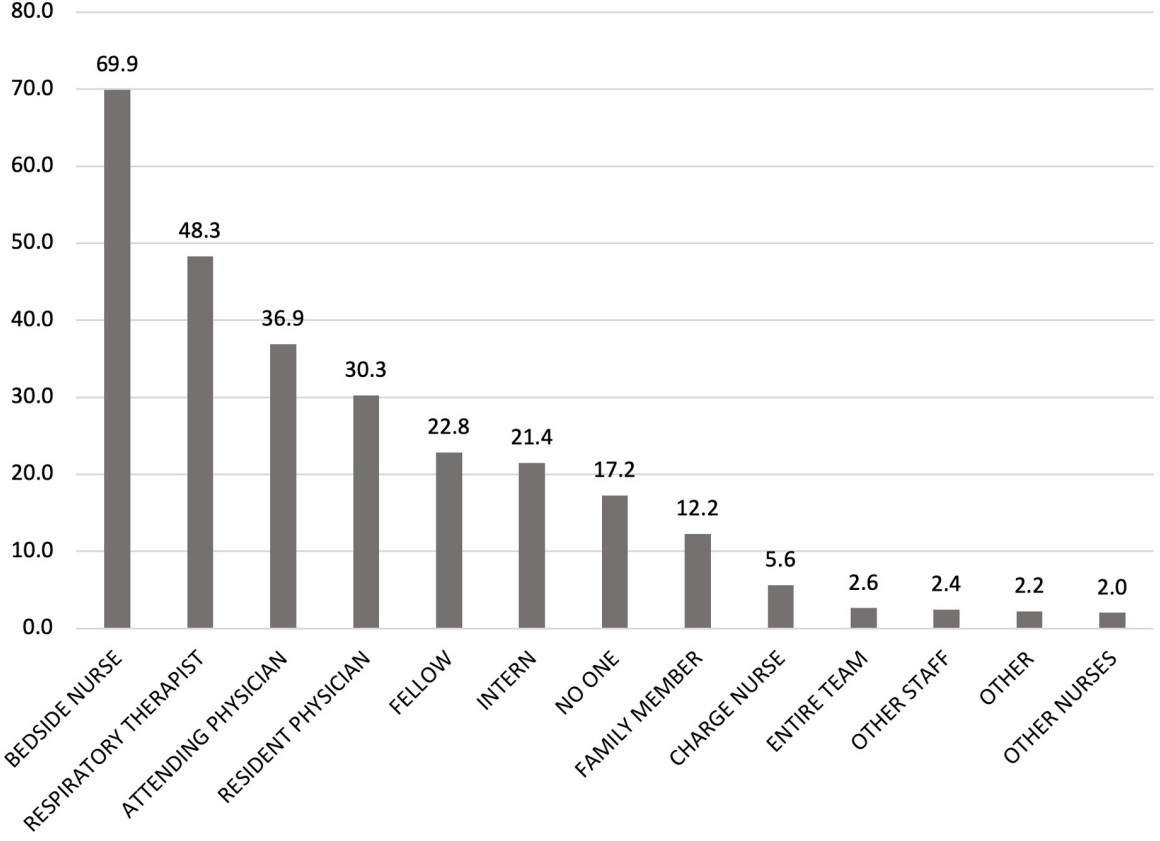

**Fig 1. Frequency of clinician interactions as reported by physicians (n = 499).**

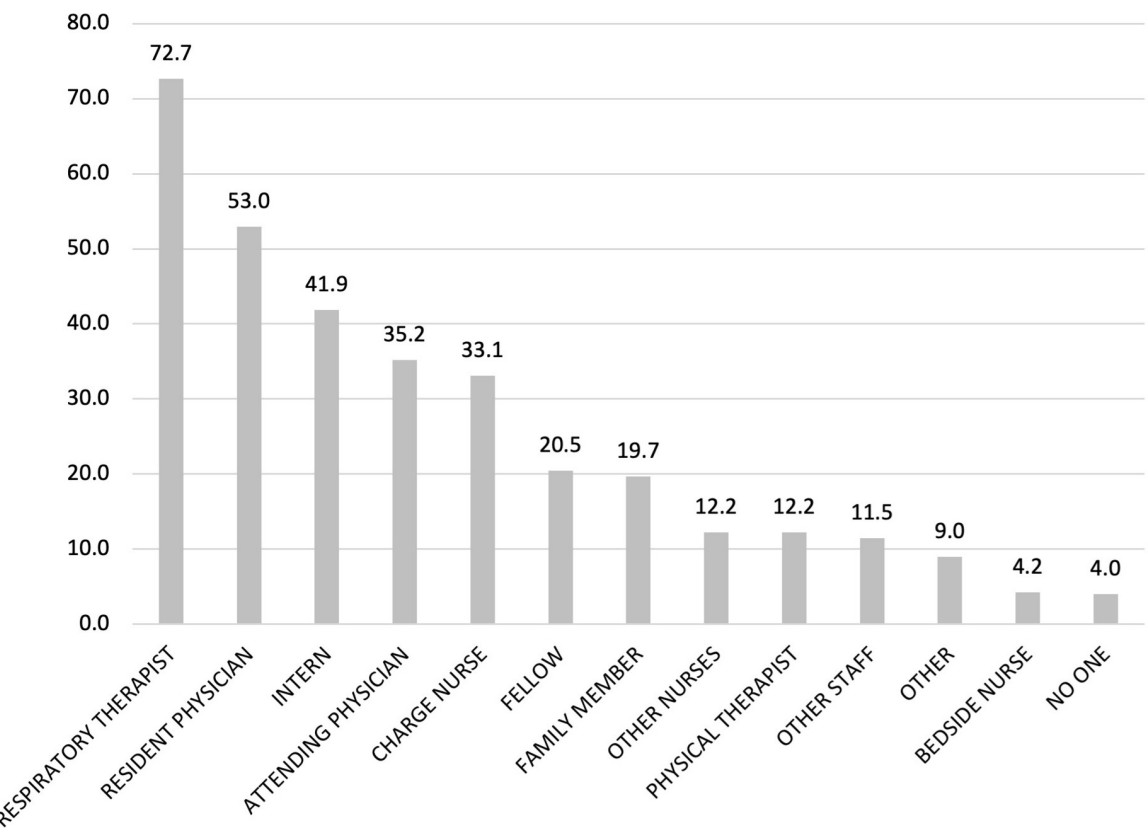

**Fig 2. Frequency of clinician interactions as reported by nurses (n = 523).**

Across all three clinician types, the bedside nurse was the most frequently reported interaction (reported in 69.9% of physician surveys and 75.9% of RT surveys) (**Figs 1–3**). Among physicians, interactions with the patient's attending were reported in 35.2% of nurses surveys and 21.3% of RT surveys whereas interactions with the resident were reported in 53% of nurses surveys and 27.2% of RT surveys. Interns were reported in 41.9% of all nurses surveys whereas fellows were mentioned less frequently (22.8% for physician surveys, 20.5% for nurses surveys, 15.3% for RT surveys). Across all surveys, the least reported interactions were with nurse managers and physical therapists, and other nurses (e.g., charge nurse, clinical nurse specialist, clinical nurse leader or other staff nurse). On the day shift, cinicians reported interacting with family members although this varied by clinician (Figs 1–3): physicians reported interacting with a family member on 18.5% of day-shift surveys (S1 Fig), nurses on 24.6% of day-shift surveys (S2 Fig), and RTs on 5.3% of day-shift surveys (S3 Fig). Interactions with family members were less frequent overnight, with only 10.8% of nurse night-shift surveys (S2 Fig) and <5% of physician and RT night-shift surveys reporting such interactions (S1 and S3 Figs).

Overall, clinicians reported interacting with one fewer individual during the night shift than during the day shift (1.99 versus 3.02 individuals) (Table 1). In addition to fewer interactions with family members, attending physician interactions were less frequently reported during night shifts: only 9.2% of RT night-shift surveys and <5% of nurses and physician night-shift surveys reported interacting with the attending physician (S1–S3 Figs). Notably, 34.8% of physician surveys, 8.6% of nurses surveys, and 19.4% of RT surveys taken during the night shift reported no interactions with any individuals over the patient's care that shift (S1–S3 Figs).

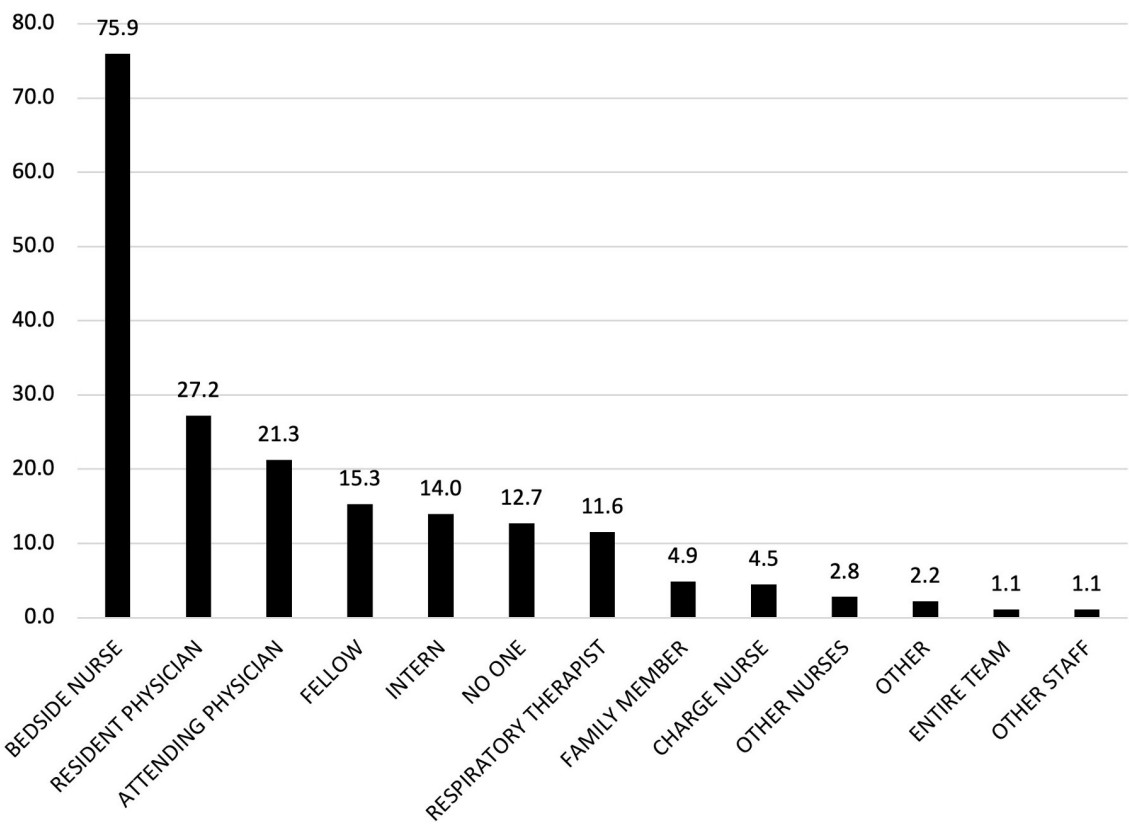

**Fig 3. Frequency of clinician interactions as reported by RTs (n = 536).**

## Discussion

In 2 ICUs in the Midwest, we used a network lens to identify a small number of interprofessional team member interactions each shift when delivering ABCDE. Nurses interacted most often with other clinicians and family members. While this may appear self-evident, until now, there was no empirical data demonstrating individual clinicians interactions in ABCDE for each patient, each shift. Given the importance of team interactions to effective teamwork [12], as well as the low penetrance of ABCDE implementation [4], such data, regarding who is interacting with whom for ABCDE, can be used to inform interventions, targeted at the patient-level ICU team, each shift, to improve team coordination, teamwork and ABCDE delivery. Compared to unit-level interventions to improve ABCDE which have had limited effectiveness [4], a patient-shift-level intervention approach is consistent with the actual care delivery and has the potential to lead to the consistent delivery of ABCDE for ICU patients. Specifically, our results suggest three possible approaches for designing patient-shift-level interventions to improve ABCDE–targeting the interprofessional shift-level team, focusing on the nurse, and examining how to include family members in ABCDE care.

First, we identified that teams were comprised of 3–4 individuals, interprofessional, change from shift-to-shift, and that nurses have the most interactions. These findings empirically underscore the importance of interprofessional ICU care [28], specifically mechanical ventilation care [1, 10, 11]. These findings are novel since previous research primarily focused on everyone in the ICU as the team [16] as the primary target for ABCDE quality improvement [4], and traditionally, considered these teams to be fixed [29, 30]. Yet our data indicates a small

number of team interactions with variability in the reported interactions that suggests interprofessional teams are dynamic. Together these data suggest that future work to improve ABCDE should focus on the interprofessional team present each shift for each patient as targets for improvement. An example of a patient-shift-level intervention could include a huddle with the nurse, physician, and RT assigned for each patient to discuss ABCDE. Huddles are a well-established approach that involves the team convening and discussing patient care in pre-scheduled intervals [31]; huddles have improved team communication and subsequently patient outcomes [32–34]. Because nurses had the most interactions within the team, nurse-led huddles could be particularly helpful. Indeed, a nurse-led huddle for early rehabilitation was effective in improving mobility for ICU patients [35].

Clinicians reported family member interactions in 25% of shifts (for nurses) and as low as 5% of surveys (for RTs); these percentages were also lower on night shift. While surprising in light of recent efforts to increase family member involvement in ICU care [36], our data indicate that more work is needed to incorporate family members in mechanical ventilation care. Targeting nurses, since 25% of nurse surveys reported family member interactions, may be an important first step in improving family member involvement. These data support various ongoing efforts to include family members in delirium monitoring [37, 38] and in direct patient care activities [39]. Understanding how to further engage families in critical care, specifically in ABCDE, may be necessary to improve ICU care.

Perhaps unexpectedly, 4–17% of surveys indicated no contact for ABCDE delivery that shift (more often on night shift), meaning none of the surveyed clinicians reported interacting with team members for ABCDE care. Additionally, a low percentage of nurses reported interacting with other nurses on night shift. These findings could reflect that teams may not interact as much overnight in general, there may be limited ABCDE activities overnight (e.g., no early mobility on night shift) or may reflect the protocolized nature of ABCDE delivery. While lack of interactions among the team may be surprising, protocolized care has been described as being integral to interprofessional collaboration [28], It is possible that protocolized evidence-based care has diminished the need for clinicians to interact, possibly explaining the lack of interactions. Additional research, using qualitative methods, to examine and contextualize how the team interacts and coordinates ABCDE delivery may be helpful.

Despite our study's strengths, we acknowledge limitations. We may not have captured all interactions because we collected data only on weekdays (Monday-Friday), and because residents (not attendings) were our target respondents. Visiting hours may have differed at our two study sites resulting in varying family member interactions. Additionally, allied health presence varied in our study ICUs (i.e., physical therapists were not present on rounds); this may explain the limited interactions reported and may impact generalizability. We did not collect demographic information, aside from clinical role and site, about our study participants and we did not survey patients; these are noted limitations. We used self-reported interactions which may be subject to recall bias. Moreover, our data does not capture the quality, degree, or frequency of each interaction. However these critiques are also applicable to EHR based approaches, and likely to an even greater extent. Our study included ICUs where ABCDE care was reported as a routine part of care, thus limiting the generalizability of our findings. Finally, our study design surveyed only the nurses, physicans, and RTs, thus perceptions of other ICU clinicians (e.g., physical therapists, pharmacists) were not included; although it is important to note that these roles could be included if the nurse, physician or RT reported contact with these clinicians in our survey. Based on these limitaitons, future research should examine team interactions in a larger sample, include patients and family members as survey repsondents and focus on understanding how to improve team member interactions, ABCDE delivery and patient care considering brief shift-level interventions.

## Conclusion

We empirically identified that ICU patient-shift team interactions were relatively small (3–4 interactions), fewer on night shift, dynamic and often interprofessional. Nurses had the most interactions within the teams. Interventions targeting patient, shift-level teams, focusing on nurses and exploring how to include family members, may be an important target for future work focused on improving ABCDE delivery in adult medical ICUs.

## Supporting information

**S1 Fig. Frequency of clinician interactions as reported by MDs, day/night (n = 499).** (TIF)

**S2 Fig. Frequency of clinician interactions as reported by RNs, day/night (n = 523).** (TIF)

**S3 Fig. Frequency of clinician interactions as reported by RTs, day/night, day/night (n = 536).** (TIF)

**S1 File.** (DOCX)

## Acknowledgments

We thank the ICUs and participants for their participation in our study.

## Author Contributions

**Conceptualization:** Deena Kelly Costa.

**Data curation:** Emily M. Boltey, Deena Kelly Costa.

**Formal analysis:** Hannah C. Ratliff, Olga Yakusheva, Emily M. Boltey.

**Funding acquisition:** Hannah C. Ratliff, Deena Kelly Costa.

**Investigation:** Olga Yakusheva, Deena Kelly Costa.

**Methodology:** Olga Yakusheva, Emily M. Boltey, Deanna J. Marriott, Deena Kelly Costa.

**Project administration:** Olga Yakusheva, Emily M. Boltey, Deena Kelly Costa.

**Resources:** Hannah C. Ratliff, Olga Yakusheva, Deena Kelly Costa.

**Software:** Deena Kelly Costa.

**Supervision:** Deanna J. Marriott, Deena Kelly Costa.

**Validation:** Deanna J. Marriott, Deena Kelly Costa.

**Visualization:** Hannah C. Ratliff, Olga Yakusheva, Deanna J. Marriott, Deena Kelly Costa.

**Writing – original draft:** Hannah C. Ratliff, Deena Kelly Costa.

**Writing – review & editing:** Hannah C. Ratliff, Olga Yakusheva, Emily M. Boltey, Deanna J. Marriott, Deena Kelly Costa.

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
