## [Decision Letter · Decision Letter 0]

25 Sep 2023

PONE-D-23-25295Patterns of interactions among ICU interprofessional teams: A prospective patient-shift-level survey approachPLOS ONE

Dear Dr. Costa,

Thank you for submitting your manuscript to PLOS ONE. After careful consideration, we feel that it has merit but does not fully meet PLOS ONE’s publication criteria as it currently stands. Therefore, we invite you to 

submit a revised version of the manuscript that addresses the points raised during the review process.

Based on Reviewers’ comments and my own assessment of your manuscript, I would like to inform you that your work doesn’t meet the criteria for publication with Plos One Journal in its current form, however, if you can address all reviewers concerns and my own comments, I can consider reviewing your revised manuscript.

Specifically, the manuscript needs to be re-written to clearly state the primary objective of the study and the main results in a consistent and non-confusing way. For example, in the abstract and results section, you state that the interaction with family members is common while it is below 20% for all 3 healthcare professionals. Please correct this unless if the standard of care is not to involve family members every time (100% of the time you do ABCDE).

In the methods section, you need to clearly define the standards of care or expected/ideal number of interactions between clinicians every shift? Also, comment on the ideal number of staff in your settings.

In your discussion, please explain if there is evidence that more interactions are associated with better care or better outcomes.

Also, make a comment about having less interactions during the night is expected or not, especially that some elements of ABCDE are not planned during the night like awakening and mobility because it is the time for sleeping for patients as well.

Please submit your revised manuscript by Nov 09 2023 11:59PM. If you will need more time than this to complete your revisions, please reply to this message or contact the journal office at plosone@plos.org. Please include the following items when submitting your revised manuscript:A rebuttal letter that responds to each point raised by the academic editor and reviewer(s). You should upload this letter as a separate file labeled 'Response to Reviewers'.A marked-up copy of your manuscript that highlights changes made to the original version. You should upload this as a separate file labeled 'Revised Manuscript with Track Changes'.An unmarked version of your revised paper without tracked changes. You should upload this as a separate file labeled 'Manuscript'.

We look forward to receiving your revised manuscript.

Kind regards,

Eugene Tuyishime, MBBS

Academic Editor

PLOS ONE

Journal Requirements:

Reviewers' comments:

Reviewer's Responses to Questions

**Comments to the Author**

1. Is the manuscript technically sound, and do the data support the conclusions?

Reviewer #1: Yes

Reviewer #2: Partly

Reviewer #3: Partly

2. Has the statistical analysis been performed appropriately and rigorously? 

Reviewer #1: Yes

Reviewer #2: Yes

Reviewer #3: Yes

3. Have the authors made all data underlying the findings in their manuscript fully available?

Reviewer #1: Yes

Reviewer #2: Yes

Reviewer #3: No

4. Is the manuscript presented in an intelligible fashion and written in standard English?

Reviewer #1: Yes

Reviewer #2: Yes

Reviewer #3: No

5. Review Comments to the Author

Reviewer #1: Thank you for the opportunity to review this manuscript. In this paper, the authors describe patterns of interactions between multidisciplinary ICU team members in 2 medical ICUs as they relate to the delivery of the ABCDE care bundle in mechanically ventilated patients. Overall, this is a very well-written and interesting paper with an appropriately focused approach and nice description of significance, methodology, results, and discussion alike. I have some brief comments that I hope will be of use to the authors and editors.

Specific comments:

Page 5, paragraph starting on line 121 – This comment about the limitations of EHRs as a data source is useful and insightful.

Page 7, line 155 – I am a little surprised that there was so little interaction with the physical therapist in either this study or in the authors’ prior work. (They seem pretty key for the “E” part of ABCDE.) Is this a function in part of how multidisciplinary ICU rounds work at these institutions? (Not to brag or anything, but I talk with the PTs about every patient every day, at least briefly, since they round with us.)

Page 8, lines 182-184 – Consider rephrasing this statement as “clinicians had the option to select up to 12 individuals by role, document their names, and list any additional clinicians if applicable” to improve clarity.

Page 8, line 198 – I am not sure what it means for data collection to occur over “12-16 weeks”. How long was it? (12 weeks in one ICU, 16 weeks in another?)

Page 8, line 203 – I assume the authors mean that a mechanically ventilated patient’s duration of mechanical ventilation is approximately 4 days, not their entire hospital length of stay?

Page 9, line 217 – “We provided candy as an incentive.” This is the first time I have seen this as a comment in a scientific paper, and I think I love it. (I propose that part of your lower participation rate for attending physicians is that we may be less susceptible to candy-based bribery.)

Page 12, line 283 – Page 13, line 306 – I would appreciate it if the authors could clarify an item for me. When they say that a clinician interacted with someone else (a family member or another clinical team member), do they mean they interacted specifically regarding ABCDE issues or that they interacted at all? It seems plausible that a night shift physician might not really check in with other team members 34.8% of the time for a stable ventilated patient, for example (since they’re mostly there overnight for acute issues). The statement that only 18.5% of day shift physicians spoke to a family member on a given day, however, strikes me as quite low. Was this a reference only to ABCDE-relevant discussions or all family discussions?

Reviewer #2: Dear authors,

thank you very much for the opportunity to review the submitted manuscript. The manuscript includes a survey about interactions of clinicians within ICU teams. I have to admit that I am usually appreciating the work by Deena Costa and her group, but this time I have difficulties to follow.

The basic idea is to count interactions within the team to assess the performance of the ABCDE bundle. This is genius, innovative and clear. The manuscript is written well, literature and tables are fine. At all, well done.

What I do not understand (or appreciate) is the approach to include MD, RN, and RT only. First, the team is not defined, and you should add such definition. Second, “E” is early exercise and mobilization, and usually performed by physical therapists. But these are not included or even mentioned here (only in section limitations). You should explain this, because future PT readers will wonder about this. “C” can be related to pharmacists, but similar, these are not mentioned. The arguments, there would be “substantial variations in ICU staffing”, or how AHP would prevent from achieving the goal of the study, are hard to follow. Your approach of team-understanding is exceptional and leads to a very limited generalizability. Take care of this.

If you reduce the ABCDE-team to MD, RN, and RT, it is not surprising that the “team is small”. And by doing this, it might be a “novel”, but it is also self-evident, as you say. Contrary, it is reasonable to “focus on everyone in the ICU as the team”, because for keeping an ICU running, you cannot only perform ABCDE bundles, you have to get supplies, clean the rooms, organize staff, and perform many other processes. MD, RN, and RT are continuously communicating and organizing different processes and might be hampered or supported in performing the ABCDE bundle. Of course, if someone is cleaning the ICU room, a physical therapist would wait respectful for this person, until the PT mobilizes a patient. This is teamplay. Hence, reducing the team to the active parts only, is misleading and does not consider real-life interaction and business. The approach by the authors is not understandable, and not practicable. You focus on individuals of three professions, but it will fail in practice because ICU teams consist of more than these three professions, and the work on ICU is more than performing the ABCDE bundle.

Families & patients: your objective is to determine the number and types of individuals, interacting in the care of MV patients, and you count the family as “individuals”. Sometimes, it seems that families are part of the team, sometimes not. Again, you should provide a clear definition. This leads to the question why you define your approach as ABCDE, but not ABCDEF? The A2F bundle would be in line with the SCCM, and rather summarize also your approach, integrating families in the team to some (undefined) extent.

The most important question for me is: why didn’t you assessed patients as individuals in the ABCDE/F concept? Aren’t patients individuals, too? They are the most important persons within the care of the ABCDE/F bundle. In the concept of patient-centred-care, all clinicians performing the ABCDE/F bundle should interact with them, and you do not recognize their value as important team members, or their personality? I am not sure if I understand your approach, or did I misunderstand the approach?

Thank you

Reviewer #3: This is an interesting approach to looking at the ABCDE bundle in practice. I have worked as a nurse in the ICU, and some of the results were unexpected. I can see why making assumptions about the team could hamper implementation efforts. However, I do have a number of minor questions/clarifications that I hope the authors find useful (and I noted a few typos). I also have concerns about overstating the implications (or at least neglecting to identify further work needed to offset the limitations).

Abstract:

Background – second sentence is somewhat confusingly worded. It would be easier to follow as two sentences.

Interpretation: In the second sentence there is a typo in “clinicians.” The terms “small” and “smaller” lack specificity and comparison – small as compared to what?

Introduction:

Lines 105-106: National as in what nation? How many hospitals or units or patients were included – in other words, was it representative? And why was it expected to increase?

Line 131: I think this should read “survey” not “surveys.”

Methods, Study Design:

Lines 149-150: The IRB statement seems tagged on here. Maybe start the paragraph with it or put it in the section with the sample.

Line 152: You have “ABDCE” instead of “ABCDE” the second time.

Lines 156-161: Some of this seems to contradict or not address your earlier gap statement that “Despite this recognition, there has been less attention, and lack of clarity, as to who is the team and the team members involved in ABCDE each shift and how those individuals interact.” It seems that you do know who the key clinicians are, but not the rest of the team, but you’re not surveying other potential team members. Perhaps you could clarify that surveyed clinicians could name any clinician role as someone they contacted or were contacted by (e.g., physical therapy)?

Methods, Survey instrument:

Line 175: Misspelling/typo of “respondents”

175-176: Are there other relevant demographics of your participants? For example, gender, age, years of experience in role, years of experience in the ICU?

Methods, Data collection:

First paragraph – some repetitiveness about times and why times were chosen.

Lines 215-217: Why didn’t you get written consent from the clinicians? Did you get a waiver of written consent from the IRB for them?

Methods, Data analysis:

Lines 221 -226 is all one sentence and confusing – I think at least some of the parentheses and brackets are in the wrong place. This would be more understandable as multiple sentences.

Line 228: I think you mean “categorized” and not “categoried.”

Line 230-231: Consider that some of those nurse roles might be more of a leadership or administrative role, thus serving a different function from the other nurse roles.

Results:

Line 246 and others: I believe to be consistent and grammatically correct, the term should be “nurse survey,” not “nurses survey”

Line 272: I’m surprised that nurses didn’t interact more often with their nurse colleagues, especially on night shift. I would have expected some joint problem solving, especially between senior and junior nurses.

Line 283: Switching between “reported” and “referenced” is a little confusing.

Line 288: I’m wondering if the community hospital and academic hospital had the same roles present on the ICUs – for example, some community hospitals might not have interns. Or the interns might function differently in the two study site ICUs.

Line 290 - 291: I don’t think you mean “clinician respondent type” here, as physical therapists were not survey respondents. Also, I’m still a little unsure at this point about what “other” nurse means. Is it a nurse not otherwise specified, or a nurse occupying the same role as the survey respondent?

Line 293-298: It’s a little difficult to interpret this, as the patient’s family wouldn’t always be available (or exist). Having said that, it would be interesting to investigate in future work why the percent is so low (I’m not sure the numbers indicate they’re an important part of the care team when even nurses don’t mention family 25% of the time).

Table 2: Clarify in the table what the P value represents, and in the text explain why you made that comparison.

Discussion:

Line 309: Small doesn’t seem like an objective term without a basis for comparison. Also, did you identify that the teams were small, or that there were a small number of people interacting? On night shift, I would assume that attending physicians are still part of the team, even if they didn’t need to be contacted, for example. So you might need to restate your definition of team to reflect it’s the interacting team members each patient-shift.

Line 313-: I think your implications need to be tempered a little bit to reflect that these data reflect only two ICUs in the Midwest. I don’t see any suggestion of what other research needs to be done (to offset your limitations) prior to designing interventions.

6. PLOS authors have the option to publish the peer review history of their article (what does this mean?). If published, this will include your full peer review and any attached files.

Reviewer #1: No

Reviewer #2: No

Reviewer #3: No

---

## [Author Response · Author response to Decision Letter 0]

20 Dec 2023

Word document uploaded with files but also pasted here below. 

Thank you for the opportunity to respond to reviewer comments on our manuscript entitled “Patterns of interactions among ICU interprofessional teams: A prospective patient-shift level survey approach”. We appreciate the thoughtful review and have substantially revised the manuscript as a result. We believe it is considerably improved as a result. Below is a point-by-point response to reviewer comments and our corresponding edits/changes to the manuscript (with pages and line numbers indicated). 

Editor comments:

1. Specifically, the manuscript needs to be re-written to clearly state the primary objective of the study and the main results in a consistent and non-confusing way. For example, in the abstract and results section, you state that the interaction with family members is common while it is below 20% for all 3 healthcare professionals. Please correct this unless if the standard of care is not to involve family members every time (100% of the time you do ABCDE).

Thank you. We have revised the abstract (lines 69-92) and introduction (132-135) to clarify the purpose/primary objective and also clarified the results (301-307). 

2. In the methods section, you need to clearly define the standards of care or expected/ideal number of interactions between clinicians every shift? Also, comment on the ideal number of staff in your settings.

One of the strengths of our paper is that it is the first to contribute information about interactions among clinicians each shift. We know of no other data in the ICU that characterizes these interactions. However, since the reported interactions focused on the ABCDE bundle, one would anticipate there would be at least 4-5 interactions for each patient – specific to each of the care bundles. One interaction regarding spontaneous awakening, another regarding spontaneous breathing trials, another regarding coordination, another regarding delirium and regarding mobility. We have added this to methods section (lines 148-162). 

Our study ICUs were both 20-bed medical units. Each unit had approximately 50-60 nurses employed, about 5-6 respiratory therapists and varying physician presence (depending on whether it was an academic medical center or the community hospital ICU). We have added this to our methods section (lines 171-177). 

3. In your discussion, please explain if there is evidence that more interactions are associated with better care or better outcomes.

We do not know whether more interactions are associated with better outcomes. However, we do know that team interactions are foundational to effective teamwork with some emerging data that interactions among team members may have important implications for patient outcomes. In our revised manuscript, we describe this in our introduction (lines 1071115) and discussion (lines 321-324) and provide corresponding references. 

4. Also, make a comment about having less interactions during the night is expected or not, especially that some elements of ABCDE are not planned during the night like awakening and mobility because it is the time for sleeping for patients as well.

Added to our discussion (lines 361-364). 

Please submit your revised manuscript by Nov 09 2023 11:59PM. 

Journal Requirements:

We have reviewed the style requirements in detail and have updated our documents and file naming to correspond to these requirements including the main body of the manuscript, title page and supporting information as described in point #3 below. 

We have confirmed that the grant information entered into the funding information in Editorial manager is the same as the information included on our title page (K08HS024552 PI Costa; University of Michigan School of Nursing Donor Funding; Sigma Theta Tau Rho Chapter Grant). 

Thank you. We have added captions to the supporting information files at the end fo the revised manuscript, updated the in-text citations to match accordingly and have appropriately named the supporting files based on the guidance linked above. 

 

Reviewer Comments to the Author

Reviewer #1: 

1. Thank you for the opportunity to review this manuscript. In this paper, the authors describe patterns of interactions between multidisciplinary ICU team members in 2 medical ICUs as they relate to the delivery of the ABCDE care bundle in mechanically ventilated patients. Overall, this is a very well-written and interesting paper with an appropriately focused approach and nice description of significance, methodology, results, and discussion alike. I have some brief comments that I hope will be of use to the authors and editors.

Thank you. 

2. Page 5, paragraph starting on line 121 – This comment about the limitations of EHRs as a data source is useful and insightful.

Thank you. Unfortunately, due to additions we made to better explain our methods and address the reviewer comments in the following points, we had to eliminate this section in the introduction. 

3. Page 7, line 155 – I am a little surprised that there was so little interaction with the physical therapist in either this study or in the authors’ prior work. (They seem pretty key for the “E” part of ABCDE.) Is this a function in part of how multidisciplinary ICU rounds work at these institutions? (Not to brag or anything, but I talk with the PTs about every patient every day, at least briefly, since they round with us.)

In both of our study ICUs, physical therapists are not present on rounds and therefore, this may partially explain the little interaction reported with a physical therapist. We added this in our limitations section (lines 382-386). 

4. Page 8, lines 182-184 – Consider rephrasing this statement as “clinicians had the option to select up to 12 individuals by role, document their names, and list any additional clinicians if applicable” to improve clarity.

Edited. See lines 196 -199. 

5. Page 8, line 198 – I am not sure what it means for data collection to occur over “12-16 weeks”. How long was it? (12 weeks in one ICU, 16 weeks in another?)

Thank you for the opportunity to clarify. Data collection occurred in ICU 1 for 12 weeks and in ICU 2 for 16 weeks. We revised this part in the manuscript to better clarify this (lines 209-213). 

6. Page 8, line 203 – I assume the authors mean that a mechanically ventilated patient’s duration of mechanical ventilation is approximately 4 days, not their entire hospital length of stay?

Yes, thank you. We have updated the text to reflect that most mechanically ventilated patients are ventilated for approximately 4 days (lines 214-215). 

7. Page 9, line 217 – “We provided candy as an incentive.” This is the first time I have seen this as a comment in a scientific paper, and I think I love it. (I propose that part of your lower participation rate for attending physicians is that we may be less susceptible to candy-based bribery.)

Thank you! The clinicians also loved the candy. Our attending physician participation may have been impacted by candy-based bribery as you indicate but we also were primarily targeting resident physicians for the physician respondents hence, possibly explaining their lack of participation as well. We have further clarified this in the manuscript (lines 219-220 and lines 380-382). 

8. Page 12, line 283 – Page 13, line 306 – I would appreciate it if the authors could clarify an item for me. When they say that a clinician interacted with someone else (a family member or another clinical team member), do they mean they interacted specifically regarding ABCDE issues or that they interacted at all? It seems plausible that a night shift physician might not really check in with other team members 34.8% of the time for a stable ventilated patient, for example (since they’re mostly there overnight for acute issues). The statement that only 18.5% of day shift physicians spoke to a family member on a given day, however, strikes me as quite low. Was this a reference only to ABCDE-relevant discussions or all family discussions?

Our survey inquired specifically about interactions clinicians had with individuals related to ABCDE and mechanical ventilation care. Our survey did not aim to capture all interactions a clinician might have with other individuals in the care of the patient. Due to the documented importance in coordination and communication in ABCDE care and our focus on mechanically ventilated patients, we focused on understanding clinician interactions with individuals (including family members) related to ABCDE care. We have better described this in the revised manuscript (lines 144-162). 

Reviewer #2: 

1. Dear authors,

thank you very much for the opportunity to review the submitted manuscript. The manuscript includes a survey about interactions of clinicians within ICU teams. I have to admit that I am usually appreciating the work by Deena Costa and her group, but this time I have difficulties to follow.

We apologize that our paper was difficult to follow and we hope that with our responses and revisions to the manuscript in response to reviewer concerns that it is more understandable now. 

2. The basic idea is to count interactions within the team to assess the performance of the ABCDE bundle. This is genius, innovative and clear. The manuscript is written well, literature and tables are fine. At all, well done.

Thank you. 

3. What I do not understand (or appreciate) is the approach to include MD, RN, and RT only. First, the team is not defined, and you should add such definition. 

We included the physician, nurse and respiratory therapist as the clinician respondent due to prior data from our team that indicated that these were the team involved in mechanical ventilation care in the ICU.1 Further, our units had considerable variability in the presence of other clinicians (i.e., pharmacists or physical therapists were not routinely present in these units). Since we were administering an in-person survey and to ensure adequate response rates as well as ensure ability to compare results across units, we needed to focus on the clinicians that were routinely present in both units for mechanically ventilated adults. Hence our decision to survey those clinicians as survey respondents. In our revised manuscript, we better describe this decision and do note that the clinician respondents had the opportunity to indicate any other clinician they interacted with, thereby not limiting the possible reported contacts to solely these 3 clinicians (lines 144 -162). 

4. Second, “E” is early exercise and mobilization, and usually performed by physical therapists. But these are not included or even mentioned here (only in section limitations). You should explain this, because future PT readers will wonder about this. “C” can be related to pharmacists, but similar, these are not mentioned. 

We have expanded to our methods (lines 144 -162) and limitations section (lines 382 – 386) to describe why we did not include physical therapists or other allied health professionals as survey respondents. 

5. The arguments, there would be “substantial variations in ICU staffing”, or how AHP would prevent from achieving the goal of the study, are hard to follow. Your approach of team-understanding is exceptional and leads to a very limited generalizability. Take care of this.

Thank you. The study ICUs had different ICU staffing models. For example, one unit had a dedicated physical therapist whereas the other did not. One unit had nurse practitioners whereas the other did not. Dieticians were not routinely present during morning rounds and pharmacist availability also varied. Due to the variability in ICU staffing patterns, it would be nearly impossible to survey the same clinician types in each ICU each shift over a sustained period unless we focused on the nurse, physician, and respiratory therapist for each patient. Moreover, without the same respondents from each unit, we would not be able to compare our results, which was the intent of this analysis. Furthermore, prior work that I had conducted in a statewide sample of Michigan ICUs indicated that the nurse, physician, and respiratory therapist were the team involved in caring for mechanically ventilated patients. 

For all these reasons, we chose to survey the nurse, physician and respiratory therapist and inquire about all the interactions they had with other individuals when providing mechanical ventilation care. Of note, the RN, MD, and RT had the opportunity to identify many other clinicians they interacted with and while we surveyed only those providers, a variety of other clinicians were identified, including physical therapists (see Figure 2). 

To better address this, we have expanded our discussion of our survey design approach in our methods (lines 144 -162) as well as expand the limitations section (lines 382 – 386). 

6. If you reduce the ABCDE-team to MD, RN, and RT, it is not surprising that the “team is small”. And by doing this, it might be a “novel”, but it is also self-evident, as you say. Contrary, it is reasonable to “focus on everyone in the ICU as the team”, because for keeping an ICU running, you cannot only perform ABCDE bundles, you have to get supplies, clean the rooms, organize staff, and perform many other processes. MD, RN, and RT are continuously communicating and organizing different processes and might be hampered or supported in performing the ABCDE bundle. Of course, if someone is cleaning the ICU room, a physical therapist would wait respectful for this person, until the PT mobilizes a patient. This is teamplay. Hence, reducing the team to the active parts only, is misleading and does not consider real-life interaction and business. The approach by the authors is not understandable, and not practicable. You focus on individuals of three professions, but it will fail in practice because ICU teams consist of more than these three professions, and the work on ICU is more than performing the ABCDE bundle.

We agree that ICU care is more than mechanical ventilation care and the ABCDE bundle and ICU care involves a variety of individuals doing many different things. However, our included patients were mechanically ventilated patients and we were specifically interested in understanding the interactions among the team members when providing ABCDE care/mechanical ventilation care. Our data provide insight into who should be targeted to receive interventions to improve ABCDE. While environment services are integral to hospital care, we do not see how they play a part in SATs, SBTs, mobility or delirium monitoring and therefore would not be the target for an intervention to improve ABCDE care. Furthermore, our clinician respondents had the ability to indicate many other clinicians that they interacted with when providing ABCDE care, not solely the RN, MD, and RT. 

To address this comment, we have clarified our findings in our discussion to highlight that we identified a small number of individuals interacting when providing ABCDE care (lines 317-319 and lines 400-402). 

7. Families & patients: your objective is to determine the number and types of individuals, interacting in the care of MV patients, and you count the family as “individuals”. Sometimes, it seems that families are part of the team, sometimes not. Again, you should provide a clear definition. This leads to the question why you define your approach as ABCDE, but not ABCDEF? The A2F bundle would be in line with the SCCM, and rather summarize also your approach, integrating families in the team to some (undefined) extent.

Our surveys included family members as possible contacts however we did not survey family members in our study. While the ABCDEF bundle supports family members as part of the team, we know of no empiric evidence that demonstrates whether they are included in the team or not. Our data, on who the clinicians are interacting with when providing ABCDE care to mechanically ventilated patients highlights that families and patients may be a part of the team. At the time of the study, there was limited data supporting the efficacy of the A2F bundle and therefore, given the evidence supporting the effectiveness of ABCDE, we opted to survey about ABCDE interactions and focus more clearly on high quality mechanical ventilation care. 

8. The most important question for me is: why didn’t you assessed patients as individuals in the ABCDE/F concept? Aren’t patients individuals, too? They are the most important persons within the care of the ABCDE/F bundle. In the concept of patient-centred-care, all clinicians performing the ABCDE/F bundle should interact with them, and you do not recognize their value as important team members, or their personality? I am not sure if I understand your approach, or did I misunderstand the approach? Thank you. 

Thank you. We agree that patients are individuals and clinicians should be interacting with them when providing care. Unfortunately, it was out of the scope of work for this study; it is a limitation of our study that we did not survey patients and inquire about who contacted them. We added to our limitations that we did not include patients as respondents (lines 384-386). 

Reviewer #3: 

1. This is an interesting approach to looking at the ABCDE bundle in practice. I have worked as a nurse in the ICU, and some of the results were unexpected. I can see why making assumptions about the team could hamper implementation efforts. However, I do have a number of minor questions/clarifications that I hope the authors find useful (and I noted a few typos). I also have concerns about overstating the implications (or at least neglecting to identify further work needed to offset the limitations).

Abstract:

1. Background – second sentence is somewhat confusingly worded. It would be easier to follow as two sentences.

 Edited (lines 68-69) 

2. Interpretation: In the second sentence there is a typo in “clinicians.” The terms “small” and “smaller” lack specificity and comparison – small as compared to what?

Thank you. We revised the abstract interpretation to better clarify our findings. (lines 89-92)

3. Introduction. Lines 105-106: National as in what nation? How many hospitals or units or patients were included – in other words, was it representative? And why was it expected to increase?

 Thank you! We clarified this (lines 97-100). 

4. Line 131: I think this should read “survey” not “surveys.”

Edited. 

5. Methods, Study Design:

a. Lines 149-150: The IRB statement seems tagged on here. Maybe start the paragraph with it or put it in the section with the sample.

Consistent with PLOS One formatting requirements, the IRB statement must be included in the methods section. However, we did move it to the end of the sample subsection within Methods (lines 186-188). 

b. Line 152: You have “ABDCE” instead of “ABCDE” the second time.

Edited. 

c. Lines 156-161: Some of this seems to contradict or not address your earlier gap statement that “Despite this recognition, there has been less attention, and lack of clarity, as to who is the team and the team members involved in ABCDE each shift and how those individuals interact.” It seems that you do know who the key clinicians are, but not the rest of the team, but you’re not surveying other potential team members. Perhaps you could clarify that surveyed clinicians could name any clinician role as someone they contacted or were contacted by (e.g., physical therapy)?

In response to this comment and those made by the other reviewers, we have added more specificity to our methods to describe that the respondents could indicate any clinician they interacted with, not solely the RN, MD, or RT respondents. See Reviewer #2, point #3 above and lines 196-199.

6. Methods, Survey instrument:

a. Line 175: Misspelling/typo of “respondents”. 

Edited. 

b. 175-176: Are there other relevant demographics of your participants? For example, gender, age, years of experience in role, years of experience in the ICU? 

Aside from clinical role and unit, we unfortunately did not collect to describe the demographics of our participant sample. We have added this as a limitation (lines 384-386). 

7. Methods, Data collection. First paragraph – some repetitiveness about times and why times were chosen.

Edited.

a. Lines 215-217: Why didn’t you get written consent from the clinicians? Did you get a waiver of written consent from the IRB for them?

To streamline data collection, we received verbal consent from clinicians. We did receive a waiver of documentation of informed consent from the IRB and we have added this to the ethics section in the methods (lines 186-188). 

8. Methods, Data analysis:

a. Lines 221 -226 is all one sentence and confusing – I think at least some of the parentheses and brackets are in the wrong place. This would be more understandable as multiple sentences.

Edited and moved to data analysis section (lines 230-233). 

b. Line 228: I think you mean “categorized” and not “categoried.”

Edited (line 241). 

c. Line 230-231: Consider that some of those nurse roles might be more of a leadership or administrative role, thus serving a different function from the other nurse roles.

Agreed. To assess interprofessional patterns of interactions we included other nurse roles under the nurse category (lines 230-231) however, in the Figures, we highlight the percentage of interactions reported with each individual role, including charge, nurse manager etc, by each clinician respondent (RN, MD or RT) to better demonstrate the individual patterns of interactions. 

Results:

d. Line 246 and others: I believe to be consistent and grammatically correct, the term should be “nurse survey,” not “nurses survey”

Edited.

e. Line 272: I’m surprised that nurses didn’t interact more often with their nurse colleagues, especially on night shift. I would have expected some joint problem solving, especially between senior and junior nurses.

We now note the lack of interaction among nurses on night shift in our discussion (lines 361-364). 

f. Line 283: Switching between “reported” and “referenced” is a little confusing.

We have edited throughout to keep terms consistent and now refer to ‘reported’ only in text and in tables. 

g. Line 288: I’m wondering if the community hospital and academic hospital had the same roles present on the ICUs – for example, some community hospitals might not have interns. Or the interns might function differently in the two study site ICUs.

We now provide additional detail about the two hospitals and the study ICUs (lines 171-178) 

h. Line 290 - 291: I don’t think you mean “clinician respondent type” here, as physical therapists were not survey respondents. Also, I’m still a little unsure at this point about what “other” nurse means. Is it a nurse not otherwise specified, or a nurse occupying the same role as the survey respondent?

Thank you. We edited this section and also described what “other nurses” means (lines 299-302). 

i. Line 293-298: It’s a little difficult to interpret this, as the patient’s family wouldn’t always be available (or exist). Having said that, it would be interesting to investigate in future work why the percent is so low (I’m not sure the numbers indicate they’re an important part of the care team when even nurses don’t mention family 25% of the time).

Thank you. We clarify this in our revised manuscript in response to your comment and other reviewers. One rationale may be that clinicians were reporting on interactions related to ABCDE care and therefore, family members may not be involved because they may not be available (or exist), as you suggest. We expand our discussion to include that future work should investigate how to include family members in mechanical ventilation care. (lines 607-615). 

j. Table 2: Clarify in the table what the P value represents, and in the text explain why you made that comparison.

We tested for statistical difference between day and night shift reported interactions in Table 2 and have added this to the text (lines 349-357). 

9. Discussion:

a. Line 309: Small doesn’t seem like an objective term without a basis for comparison. Also, did you identify that the teams were small, or that there were a small number of people interacting? On night shift, I would assume that attending physicians are still part of the team, even if they didn’t need to be contacted, for example. So you might need to restate your definition of team to reflect it’s the interacting team members each patient-shift.

Thank you. This is an excellent suggestion. We have revised our discussion to highlight that our study demonstrates a small number of interprofessional team member interactions each shift when delivering ABCDE (lines 317-318 and 400-401). 

b. Line 313-: I think your implications need to be tempered a little bit to reflect that these data reflect only two ICUs in the Midwest. I don’t see any suggestion of what other research needs to be done (to offset your limitations) prior to designing interventions.

We have revised our discussion to reflect that our data are from 2 ICUs in the Midwest only (lines 317-318) and have tempered the discussion to reflect the sample size as well as expanded the limitations section (lines 379-394). In addition to sentences at the end of each paragraph in the discussion describing future research, we added a concluding summary of next steps in the discussion (lines 394-397).

---

## [Editor Report · Decision Letter 1]

29 Jan 2024

Patterns of interactions among ICU interprofessional teams: A prospective patient-shift-level survey approach

PONE-D-23-25295R1

Dear Deena Kelly Costa,

We’re pleased to inform you that your manuscript has been judged scientifically suitable for publication and will be formally accepted for publication once it meets all outstanding technical requirements.

Kind regards,

Eugene Tuyishime, MBBS, MMed, MSc

Academic Editor

PLOS ONE

Additional Editor Comments:

We appreciate you submitting your manuscript to Plos One and hope you will consider us again for future submissions.

---

## [Editor Report · Acceptance letter]

1 Apr 2024

PONE-D-23-25295R1 

PLOS ONE

Dear Dr. Costa, 

I'm pleased to inform you that your manuscript has been deemed suitable for publication in PLOS ONE. Congratulations! Your manuscript is now being handed over to our production team.

Kind regards, 

on behalf of

Dr. Eugene Tuyishime 

Academic Editor

PLOS ONE